# Metabolic and Biochemical Effects of Pyrroloquinoline Quinone (PQQ) on Inflammation and Mitochondrial Dysfunction: Potential Health Benefits in Obesity and Future Perspectives

**DOI:** 10.3390/antiox13091027

**Published:** 2024-08-24

**Authors:** Davide Charrier, Giuseppe Cerullo, Roberta Carpenito, Vincenzo Vindigni, Franco Bassetto, Luca Simoni, Tatiana Moro, Antonio Paoli

**Affiliations:** 1Department of Biomedical Sciences, University of Padova, 35122 Padua, Italy; davide.charrier@phd.unipd.it (D.C.); luca.simoni@phd.unipd.it (L.S.); tatiana.moro@unipd.it (T.M.); antonio.paoli@unipd.it (A.P.); 2Plastic and Reconstructive Surgery Unit, Department of Neurosciences, University of Padua, 35122 Padua, Italyvincenzo.vindigni@unipd.it (V.V.); franco.bassetto@unipd.it (F.B.); 3Research Center for High Performance Sport, UCAM Catholic University of Murcia, 30107 Murcia, Spain

**Keywords:** pyrroloquinoline quinone (PQQ), adipose tissue, oxidative stress, inflammation, mitochondrial dysfunction

## Abstract

Obesity is defined as a complex, systemic disease characterized by excessive and dysfunctional adipose tissue, leading to adverse health effects. This condition is marked by low-grade inflammation, oxidative stress, and metabolic abnormalities, including mitochondrial dysfunction. These factors promote energy dysregulation and impact body composition not only by increasing body fat but also by promoting skeletal muscle mass atrophy. The decline in muscle mass is associated with an increased risk of all-cause mortality in individuals with this disease. The European Food Safety Authority approved pyrroloquinoline quinone (PQQ), a natural compound, as a dietary supplement in 2018. This narrative review aims to provide a comprehensive overview of the potential role of PQQ, based on its anti-inflammatory and antioxidant properties, in addressing dysfunctional adipose tissue metabolism and related disorders.

## 1. Introduction

Obesity is recognized by the World Health Organization (WHO) as one of the most severe public health concerns; it is defined as an excessive accumulation of adipose tissue [1]. Although obesity is typically seen as the consequence of an energy imbalance between calorie consumption and expenditure, it is actually a multifactorial, complex chronic disease also characterized by increased oxidative stress and a low-grade inflammation status [2].

Indeed, the Krebs cycle and the mitochondrial respiratory chain can be overloaded by excessive energy intake, which is related to overeating behaviors. This energy intake overload leads to mitochondrial dysfunction and to an increase in the generation of reactive oxygen species (ROS). This increase in ROS generation by the respiratory chain exacerbates oxidative stress, further amplifying the inflammatory process. Enhanced low-grade inflammation and oxidative stress, in turn, promote metabolic abnormalities, including mitochondrial dysfunction [3]. The imbalance between energy intake and expenditure leads also, obviously, to an increase in adipose tissue. This condition results in several consequences.

Various physiological processes, such as energy regulation and inflammation, are affected by an excessive amount of adipose tissue, which also influences the function of other organs, as described by other authors [4]. Abnormal adipose tissue amount and/or physiology, oxidative stress, inflammation, and mitochondrial dysfunction play crucial roles in regulating skeletal muscle mass [5,6,7]. These factors promote energy dysregulation and interfere with skeletal muscle remodeling by enhancing catabolic processes [8]. A decline in muscle mass, combined with increased fat mass, leads to metabolic inflexibility, ectopic lipid accumulation, and the formation of toxic lipid intermediates [9].

Without appropriate interventions, all these conditions may contribute to the development of a variety of diseases, such as cardiovascular disease, diabetes, and cancer [10].

Among the many proposed interventions, nutritional supplements have been advocated as useful for improving fat tissue metabolism.

Pyrroloquinoline quinone (PQQ) is a compound that exists naturally in certain foods and is available as a dietary supplement in its disodium crystal state. In 2018, the European Commission approved PQQ as a novel food ingredient known as MGCPQQ^®^, after a positive report by the European Food Safety Authority [11]. In particular, the antioxidant and anti-inflammatory properties of PQQ have been deeply investigated in relation to its potential health benefits [12,13,14]. Numerous in vivo studies have demonstrated PQQ’s ability to reduce body fat accumulation [15], mitigate oxidative stress [16], improve mitochondrial health [17], and counteract inflammation [18]. Recently, additional studies have shown PQQ’s potential role in muscle health, even in humans [19,20,21,22,23].

The objective of this narrative review is to provide a comprehensive overview of the potential therapeutic uses of PQQ, focusing on its anti-inflammatory and antioxidant properties, against dysfunctional adipose tissue metabolism.

## 2. PQQ: Structure and General Characteristics

PQQ is an aromatic tricyclic o-quinone that was originally discovered as a cofactor for bacterial dehydrogenases [24]. Its chemical structure comprises 4,5-dihydro-4,5-dioxo-1H-pyrrolo [2,3-f] quinoline-2,7,9-tricarboxylic acid, existing in a redox-active o-quinone form, which can be reversibly reduced to pyrroloquinoline quinol through a semiquinone intermediate [25]. In nature, PQQ can be found in a variety of dietary sources, such as fermented soybeans, tea, fruits, vegetables, and human milk, with concentrations ranging from 3.7 to 50 ng/g [26]. The highest PQQ content has been found in fermented soybeans, at 61 ng/g. In the human diet, the daily intake of PQQ and its derivatives has been estimated to range between 0.1 and 2 mg [27,28].

Following ingestion, PQQ is rapidly absorbed in the intestine, with peak blood levels occurring within 2–3 h [29]. Smidt et al. found that in mice, at least 62% (range 19–89%) of the consumed amount is absorbed within 24 h, after which it decreases to low levels [29]. Indeed, roughly 80% of the amount absorbed is excreted in the urine after 24 h, while around 20% is retained within the tissues, mostly in the skin and kidney [28,30]. In healthy subjects, the concentrations of PQQ in plasma and urine were found to be 1.7 ± 0.6 ng/mL and 0.8 ± 0.2 ng/mL, respectively [31]. In the same study, the analysis of human tissues obtained from cadavers revealed trace amounts of PQQ in several organs, including the liver, intestine, kidney, lung, and pancreas. The highest level of PQQ was observed in the spleen (5.9 ± 3.4 ng/g wet tissue) [31]. 

Due to its structure, PQQ plays a pivotal role in many cellular processes, functioning as an antioxidant, a cofactor, and a regulator of numerous cellular pathways (Figure 1) [12].

As an antioxidant, PQQ can act as a radical scavenger on aroxyl (ArO(•)) and peroxyl (ROO(•)) radicals, thereby preserving cells from oxidative damage [13]. In comparative redox cycling assays, PQQ demonstrates an efficiency that outperforms other quinonic biofactors by 100–1000 folds [32]. Moreover, while other quinone biofactors are susceptible to auto-oxidation or condensation into inactive forms, PQQ shows remarkable stability and does not polymerize, effectively maintaining its active form [32,33,34].

As a cofactor, PQQ catalyzes several reactions involving energy metabolism. For instance, it acts as a cofactor of mammalian lactate dehydrogenase, enhancing NADH oxidation to NAD^+^ and pyruvate production [35,36].

Furthermore, PQQ may influence multiple cellular pathways, primarily by stimulating mitochondrial biogenesis, fat metabolism, and modulating the inflammatory response [12,37]. Studies have demonstrated that PQQ enhances the activation of key regulators involved in mitochondrial biogenesis and fat metabolism, such as peroxisome proliferator-activated receptors alpha and gamma (PPAR-α, PPAR-γ) and peroxisome proliferator-activated receptor gamma coactivator 1-alpha (PGC-1α) [38,39]. PQQ can also impact mitochondrial health by modulating cellular-stress-related pathways such as Janus kinase (JAK) and mitogen-activated protein kinase (MAPK), which contribute to the regulation of the immune response, mitosis, apoptosis, and cellular proliferation [40]. Moreover, PQQ can modulate inflammatory pathways by interacting with transcription factors such as nuclear factor kappa-light-chain-enhancer of activated B cells (NF-kB), which promotes the transcription of proinflammatory interleukins [41].

Given the multiple effects attributed to PQQ, its supplementation could be beneficial in humans. Long-term intake appears safe and may increase baseline PQQ concentrations, though individual absorption rates vary considerably [27].

The European Food Safety Authority (EFSA) identified a safe supplementation dosage of 20 mg per day, which is at least 250 times higher than the typical dietary intake, and established a No Observed Adverse Effect Level (NOAEL) of 100 mg/kg body mass per day [11,42]. Considering that the majority of available data come from animal studies with a daily dose of 20 mg/kg, using the calculation based on body surface area proposed by Nair and Jacob in 2016 [43], we can hypothesize that the human equivalent dose could be 1.6 mg/kg, or 112 mg per 70 kg for an adult.

## 3. PQQ and Adipose Tissue

In the last 50 years, obesity has become an epidemic and a worldwide public health emergency, significantly impacting the well-being, life expectancy, and quality of life of almost 2 billion individuals [44]. One of the main features of obesity is the increases in both visceral and subcutaneous adipose tissue [45]. While adipose tissue typically represents 15 to 28% of body mass in the general population, this percentage can exceed 40% in individuals with severe obesity [44].

However, adipose tissue not only serves as passive lipid storage but also plays an active role in regulating energy status and metabolism by producing a variety of cell-signaling molecules, such as adipokines, lipokines, and biologically active lipids [46,47]. Thus, dysfunction of adipose tissue can be associated with several metabolic disorders, including dyslipidemia and hyperglycemia, contributing to the complexity and severity of obesity [46,48].

Promising results from in vivo and in vitro studies seem to show a positive effect of PQQ on fat metabolism and body fat mass. For example, in obese mice (*C57BL/6J*), PQQ supplementation (20 mg/kg/day for 6 weeks) decreased total body and visceral fat compared to the control groups. In addition, analysis at the cellular level showed that the reduction in body fat was associated with decreased lipid content and smaller fat droplet size in adipocytes in the 3T3-L1 cell line [49]. Similarly, 16-week PQQ supplementation (20 mg/kg/day) was found to counteract the effects of benzyl butyl phthalate, an endocrine-disrupting chemical that promotes obesity and diabetes, preventing liver fat accumulation and restoring normal liver weight in *C57BL/6J* male mice [50]. PQQ supplementation reduced body mass and improved several metabolic markers including fasting blood glucose, triglycerides, total and LDL cholesterol, and insulin intolerance. These results suggest that PQQ may have the ability to mitigate various abnormal metabolic effects typically associated with obesity [50].

According to several authors, PQQ supplementation may exhibit a synergistic effect with other pharmacological compounds. For instance, PQQ supplementation alone (10 and 20 mg/kg) and in combination with atorvastatin, a commonly used statin, leads to a significant reduction in body mass gain and to an improvement in other anthropometric parameters, along with improvements in glucose homeostasis and lipid profiles. In more depth, histologic evaluation of liver tissue showed improvement in liver tissue architecture with a few infiltrations of inflammatory cells after 5 weeks of PQQ supplementation in *Sprague-Dawley* rats subjected to a high-fat, high-calorie diet designed to induce obesity [51].

From a molecular perspective, the effects of PQQ on body fat and fat metabolism might be dependent on its capacity to act at different levels. Firstly, PQQ promotes the activation of AMP-activated protein kinase (AMPK), a key regulator of energy homeo-stasis, along with its upstream regulator liver kinase B1 (LKB1) and several downstream proteins [38]. Secondly, PQQ appears to reduce the activity of key enzymes involved in fatty acid synthesis, such as acetyl-CoA carboxylase (ACC) and sterol regulatory element-binding protein 1 (SREBP-1) [52]. Thirdly, PQQ enhances the expression of PGC-1α, a master regulator of mitochondrial biogenesis and fatty acid metabolism [22,38]. This upregulation leads to enhanced fatty acid oxidation and, in general, oxidative capacity [49,53].

In summary, at least in animal models, PQQ seems to reduce body fat accumulation by (1) suppressing lipogenesis, (2) promoting fatty acid oxidation, and (3) enhancing mitochondrial biogenesis.

Regarding human studies, recently, one study evaluated the impact of 20 mg/day PQQ disodium salt supplementation on serum triglyceride and cholesterol levels in humans after 6 and 12 weeks. No significant changes were observed in mean serum TG as well as total and LDL cholesterol levels after 12 weeks of treatment with PQQ. However, a nonsignificant decrease in mean LDL cholesterol levels (from 136.1 to 127.0 mg/dL) was observed in the PQQ group [54]. Furthermore, in the stratification analysis of the high LDL cholesterol subgroup (baseline LDL cholesterol level ≥140 mg/dL), the average LDL cholesterol levels decreased significantly from the baseline in the PQQ group, indicating a potential effect of PQQ on hypercholesterolemia.

Nevertheless, research is needed to confirm these findings and explore the broader implications of PQQ supplementation for improvements in body composition and lipid profile in humans.

It is crucial to note that the mechanisms through which PQQ can enter adipocytes and impact adipose tissue metabolism are still unclear. In this regard, future investigations should focus on PQQ pharmacokinetics and on the mechanisms of tissue transportation.

## 4. PQQ and Mitochondrial Dysfunction

Obesity causes a range of cellular and metabolic alterations, such as reduced mitochondrial activity [45]. Mitochondria are the powerhouses of the cell, responsible for producing ATP through oxidative phosphorylation [46]. Dysfunctional mitochondria are less efficient at oxidizing fatty acids, leading to the accumulation of fat in tissues and contributing to obesity [55]. Moreover, mitochondrial dysfunction often results in increased production of reactive oxygen species (ROS), which can cause oxidative damage to cells and tissues, promoting inflammation and insulin resistance [56,57]. Improving mitochondrial function through lifestyle changes such as physical activity and diet can help mitigate the effects of obesity and improve metabolic health. In this scenario, the use of specific supplements such as PQQ may play a positive role.

Indeed, there are some clues suggesting that PQQ supplementation may have a role in restoring obesity-induced mitochondrial function; for example, the deprivation of PQQ leads to the suppression of mitochondrial function. Nevertheless, this suppression was reversed when 200–300 μg of PQQ per kilogram of diet was added [58]. Moreover, it was observed that adding PQQ (at doses of 10–30 µM for 24–48 h) to mouse Hepa1–6 cells led to an increase in mitochondrial DNA content and enhanced the activity of key enzymes in the Krebs cycle and oxidative phosphorylation [38]. Furthermore, 5-week PQQ supplementation (20 mg/kg/day), per se or combined with atorvastatin, enhanced hepatic mitochondrial gene expression and mtDNA content in obese *Sprague-Dawley* rats [51]. In a model of TNF-α-induced mitochondrial damage, PQQ supplementation reduced the mitochondrial damage in chondrocytes [59]. In this case, chondrocytes isolated from *C57BL/6* mice were exposed to TNF-α (50 ng/mL) or TNF-α + PQQ (10 µmol/L) for 24 h. These findings demonstrated that PQQ supplementation reduced mitochondrial damage by protecting mtDNA integrity, maintaining ATP levels, restoring mitochondrial membrane potential, and increasing the number of mitochondria [59].

PQQ appears to mitigate mitochondrial dysfunction mainly through the upregulation of PGC-1α and the reduction in ROS production. Specifically, PQQ exposure stimulates the phosphorylation of cAMP response element-binding protein (CREB) at serine 133, which activates the promoter of PGC-1α [60]. This leads to increases in both PGC-1α mRNA and protein expressions [60]. Moreover, PQQ may have a role in mitochondrial health by preventing the impairment of mitochondrial Ca^2+^ homeostasis and preserving the mitochondrial membrane potential [60].

The findings from other studies suggest a role of PQQ in controlling mitochondrial function and structural properties by inducing the transcription of genes related to cellular stress and apoptosis such as thioredoxin and MAPK [40,61]. According to Tchaparian and colleagues, PQQ may downregulate MAPK14, thereby reducing mitochondrial apoptosis, while simultaneously upregulating MAPKKK12 and PGC-1α activation, which increases mitochondriogenesis [40]. It has also been reported that PQQ can activate the JAK/STAT3 pathway, which regulates the mitochondrial electron transport chain and mitochondrial assembly [40,61].

In 2013, Harris and colleagues were among the first to link the effects of PQQ in animals to corresponding effects in humans [28]. According to their findings, PQQ supplementation (at a daily dose of 0.3 mg/kg body mass) can affect mitochondrial activity after just three days [28]. This conclusion was substantiated by alterations in the blood lactate-to-pyruvate ratio and the profile of urine metabolites, including several metabolites of the Krebs cycle, which can indirectly indicate alterations in oxidative metabolism.

Recently, promising results have also been observed by combining PQQ supplementation with exercise. In more depth, six weeks of PQQ supplementation (20 mg/day) combined with a supervised endurance exercise training program led to an augmentation in the PGC-1α muscle protein content in twenty-three non-endurance-trained men [39,62].

The results above suggest that PQQ may positively impact mitochondrial function in humans. However, the current findings are based on limited sample sizes [28], either in short-term studies [28] or specific populations (young subjects or athletes) [39]. Based on this, larger and longer-term randomized controlled trials are required, including those involving conditions typically linked to mitochondrial dysfunction, such as obesity or aging.

## 5. PQQ and Inflammation

In obesity, an excess of white adipose tissue causes local and systemic low-grade chronic inflammation. It is well known that increased secretion of tumor necrosis factor α (TNF-α), interleukin 1β (IL-1β), and interleukin 6 (IL-6), along with other proinflammatory cytokines, is common in the presence of excessive adipose tissue [44,45]. All these inflammatory factors are associated with the development of many obesity-related metabolic diseases [46]. It has been suggested that PQQ may impact obesity-related low-grade inflammation due to its antioxidant and anti-inflammatory properties. For instance, PQQ alone and with atorvastatin reduced inflammatory cytokines IL-1β, TNF-α, IL-18, and IL-6 after 5 weeks of supplementation in obese mice, resulting in improved glucose tolerance, lipid profile, and insulin sensitivity [51]. It could be speculated that the effects on glucose and insulin sensitivity are mediated by the inactivation of insulin/IGF1 signaling pathway-related genes, as reported in nematodes [63]. Furthermore, PQQ supplementation restored normal organ weight (liver and spleen) and reduced immune cell infiltration [51]. In a model of exercise-induced muscle damage, PQQ diminished the overexpression of NF-κB, TNF-α, and IL-1β [64].

PQQ exhibits anti-inflammatory properties that appear beneficial for addressing obesity-related complications, including metabolic-dysfunction-associated fatty liver disease (MAFLD), kidney dysfunction, cardiovascular disease, and neurological impairments (Figure 2).

In septic rats, PQQ supplementation alleviated acute liver injury and the apoptosis of liver cells [16]. In this study, sepsis was induced in *Sprague-Dawley* rats via cecal ligation and puncture (CLP) surgery, and PQQ (10 mg/kg) was intraperitoneally administered one hour prior to the surgery and continuously for two weeks after surgery. The results indicated that PQQ administration partially reduced the levels of inflammatory factors, including IL-6, IL-1β, and TNF-α, in both serum and liver tissues [16].

It has been proposed that PQQ can mitigate inflammatory and oxidative liver cell damage by downregulating CUL3, a regulatory protein involved in metabolism, antioxidant response, detoxification, cell proliferation/apoptosis in Kupffer cells, which are hepatic liver macrophages [16,44]. These findings suggest that PQQ supplementation could reduce the infiltration of inflammatory and immune cells, potentially decreasing the progression of liver fibrosis [65].

PQQ also shows potential in preventing kidney dysfunction, another obesity-related complication. In HK-2 kidney cells, PQQ (100 nM) modulates inflammation and senescence by reducing reactive oxygen species (ROS) production and influencing the Keap1/Nrf2 pathway, a crucial antioxidant defense mechanism in cells [66]. In particular, PQQ promotes the nuclear transcription of nuclear factor erythroid 2-related factor 2 (Nrf2), a key modulator of redox balance and signaling, which plays an important role in defending against oxidative stress damage and regulating antioxidant genes [66]. Moreover, PQQ may inhibit the activation of the nucleoside-binding domain NOD-like receptor protein 3 (NLRP3) pathway, which triggers the assembly of the inflammasome complex, leading to the maturation and secretion of proinflammatory cytokines [67].

Regarding cardiovascular health, PQQ (10 μM) seems to confer resistance to acute oxidative stress in isolated cardiomyocytes by preventing mitochondrial membrane potential depolarization [68,69]. An in vivo study further indicates that, probably due to its antioxidant effect, PQQ can reduce myocardial infarct size and improve cardiac function in rat models of ischemia [70]. A correlation between PQQ dosage (10–15–20 mg/kg) and reduction in infarct size was observed, suggesting a dose–response relationship [70].

Finally, the role of PQQ in neurological health is also noteworthy. In a mouse model of rotenone-induced neuroinflammation, PQQ pretreatment (10 μM) was shown to mitigate the inflammatory response in rotenone-treated BV2 cells through the regulation of the PINK1/Parkin-mediated pathway, thereby enhancing mitophagy [71]. To the best of our knowledge, only one study in humans indicated that oral PQQ supplementation at a daily dose of 0.3 mg/kg for three days reduced inflammatory markers like CRP and IL-6, potentially decreasing the chronic inflammatory states associated with obesity [28]. Further research is strongly recommended, focusing on large-scale, long-term clinical trials to investigate the effects of PQQ supplementation on inflammatory markers and, consequently, obesity-related diseases in humans.

## 6. PQQ and Skeletal Muscle Health

It is well established that skeletal muscle plays a fundamental role in a good state of health. Indeed, maintaining a healthy skeletal muscle mass is associated with a lower risk of mortality [72,73]. Skeletal muscle is continuously undergoing remodeling processes, mainly in response to dietary habits and mechanical and metabolic stimuli (e.g., exercise or disuse) [74]. In this regard, skeletal muscle mass homeostasis essentially depends on the balance between muscle protein synthesis and muscle protein breakdown.

Although protein and calorie intakes have substantial effects on the regulation of skeletal muscle mass, various variables should be considered in maintaining a healthy muscle environment [75,76]. As previously mentioned, obesity is strictly linked to inflammation, oxidative stress, and mitochondrial dysfunction; all these factors generally blunt the protein synthesis rate and, in parallel, enhance protein breakdown [77,78]. Promising results, especially in animal models of skeletal muscle atrophy, suggest that PQQ may have a protective role in the maintenance of skeletal muscle mass and function.

Many proinflammatory factors, such as IL-6, IL-1β, TNF-α, and NF-κB, are recognized as triggers of proteolysis and, in general, muscle loss [74,78].

PQQ administration (5 mg/kg/d intraperitoneally injected for 14 days) decreased the levels of proinflammatory cytokines (IL-6, IL-1β, and TNF-α) and reduced skeletal muscle atrophy through suppressing the JAK2/STAT3, TGF-β1/Smad3, JNK/p38 MAPK, and NF-κB signaling pathways in denervated mice [79].

Though chronic inflammation is considered a key factor leading to the loss of skeletal muscle, excessive ROS production has a crucial role in inflammation-induced skeletal muscle dysfunction [6]. Normally, skeletal muscle generates oxidant species (ROS and RONS), which work as signaling molecules in the processes of force transmission, mitochondrial biogenesis during exercise, and, in general, skeletal muscle adaptation [80,81]. When antioxidant mechanisms are unable to effectively regulate high ROS levels, the ensuing oxidative stress has a detrimental effect on the rates of protein synthesis, concurrently stimulating protein degradation. Oxidative-stress-induced muscle atrophy is strictly connected to the activation of two major proteolytic systems: the ubiquitin-proteasome system (UPS) and the autophagy-lysosome pathway (ALP) [80].

According to many authors, PQQ administration counteracts muscle atrophy mainly due to its strong antioxidant capacity. In myotubes treated with TNF-α to enhance ROS generation and induce muscle wasting, the addition of PQQ (80 µM for 24 h) significantly alleviated the reduction in myotube size by increasing their diameters, along with decreased ROS, muscle atrophy F-box (MAFbx) and muscle-specific RING finger-1 (MuRF-1) levels [23]. MuRF1 and MAFbx are two muscle-specific E3 ubiquitin ligases of the UPS [82]. The ability of PQQ to interfere with muscle atrophy by increasing the levels of atrophy-related proteins such as MuRF1 and MAFbx has also been supported by other authors [21,79].

Muscle mass homeostasis is also linked to mitochondrial activity. Indeed, as described by Romanello and Sandri, mitochondrial dysfunction is one of the most critical mechanisms of muscle atrophy [7]. 

For example, PGC-1α, a master regulator of mitochondrial biogenesis, may play a critical role in skeletal muscle adaptation by promoting fiber-type switching from glycolytic to oxidative fibers [7]. The overexpression of PGC-1α counteracts muscle mass wasting in transgenic mice subjected to denervation or fasting [83]. As shown by Kuo and colleagues in a preclinical in vivo study, the administration of PQQ (4.5 mg/kg/d subcutaneously injected for 14 days) resulted in an increase in the PGC-1 α protein level and slowed protein degradation and muscle loss [22]. A recent study also demonstrated that PQQ supplementation (20 mg/kg/day for 10 weeks) attenuated age-related muscle atrophy by increasing fiber size, mitigating chronic inflammation, and improving the redox state in the skeletal muscle of aged mice [20].

PQQ supplementation seems to affect physical function and muscle strength in humans [19]. Indeed, 12 weeks of PQQ supplementation (21.5 mg/day as PQQ disodium salt) was associated with improvements in lower limb extension muscle strength and grip strength, together with enhanced physical function (10 m shuttle walking, 6 min walking, and 10 m walking tests) in healthy Japanese volunteers.

Taken together, PQQ supplementation may have a role in muscle plasticity, especially in reducing muscle catabolism. Thus, considering that obesity is often linked to low muscle mass and physical function (sarcopenia), PQQ supplementation might improve muscle health in these individuals (Figure 3). Therefore, future studies exploring the relationship between PQQ and muscle health could potentially lead to optimized therapeutic interventions to prevent skeletal muscle wasting, given that the exact mechanisms are still unknown.

At this stage, the majority of available data come from animal models of muscle atrophy caused by denervation. More randomized controlled trials should be performed in humans before supporting its consumption to counteract the muscle wasting promoted by obesity.

## 7. Conclusions

PQQ may have strong potential as a therapeutic nutraceutical due to its antioxidant and anti-inflammatory activity, as observed in in vitro and in vivo studies. Furthermore, PQQ is associated with the attenuation of several relevant metabolic dysfunctions, such as MAFLD, glucose intolerance, lipotoxicity, ischemia, and muscle wasting. However, to the best of our knowledge, human evidence remains limited, necessitating more high-quality, randomized controlled trials.

Given that clinical studies have shown PQQ to be safe at doses up to 100 mg/day, future research might consider using doses higher than the commonly used 20 mg/day in human trials.

Further research should be performed in order to clarify (1) the kinetics of PQQ when ingested as a food ingredient; (2) the optimal dose and timing of consumption, keeping in mind that PQQ is generally safe, and the dosage used in human clinical trials is very low compared with animal models; and (3) the effects in relation to other well-known antioxidant compounds. Moreover, exploring the potential synergistic effects of PQQ with other lifestyle interventions, such as diet and exercise, could also provide valuable insights into its role in promoting health.

## Figures and Tables

**Figure 1 antioxidants-13-01027-f001:**
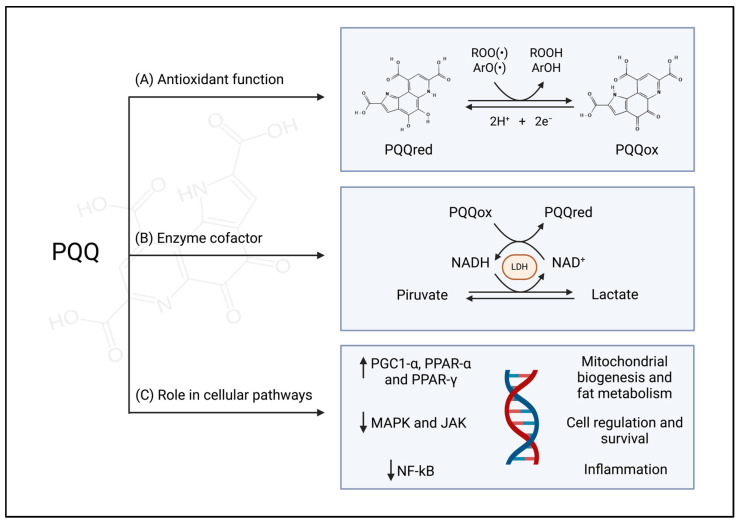
General properties of PQQ: (**A**) Antioxidant function: PQQ in its reduced state acts as a potent antioxidant, efficiently catalyzing electron transfer reactions. (**B**) Enzyme cofactor: PQQ serves as an enzyme cofactor in lactate dehydrogenase. (**C**) Role in cellular pathways: PQQ influences mitochondrial biogenesis and fat metabolism (by regulating PGC-1α, PPAR-α, and PPAR-γ), cell regulation (by regulating MAPK and JAK pathways), and inflammation (through the modulation of NF-κB). (ArO(•)): aroxyl radicals; NAD^+^/NADH: nicotinamide adenine dinucleotide oxidized and reduced; JAK: Janus kinase; LDH: Lactate dehydrogenase; MAPK: mitogen-activated protein kinase; NF-κB: nuclear factor kappa-light-chain-enhancer of activated B cells; PGC-1α: proliferator-activated receptor gamma coactivator 1-alpha; PPAR-α and PPAR-γ: Peroxisome proliferator-activated receptors alpha and gamma; PQQox: pyrroloquinoline quinone; PQQred: pyrroloquinoline quinol; (ROO(•)):Peroxyl radicals. Created with BioRender.com, accessed on 10 August 2024.

**Figure 2 antioxidants-13-01027-f002:**
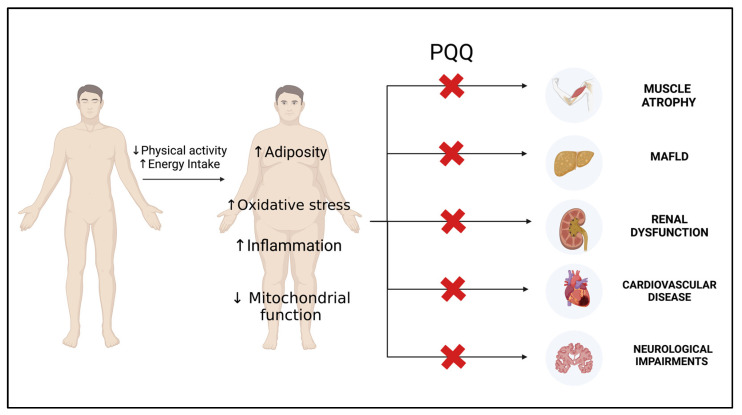
Different potential areas of action of PQQ as a protective factor against obesity-related diseases. MAFLD, metabolic-dysfunction-associated fatty liver disease. Created with BioRender.com, accessed on 31 July 2024.

**Figure 3 antioxidants-13-01027-f003:**
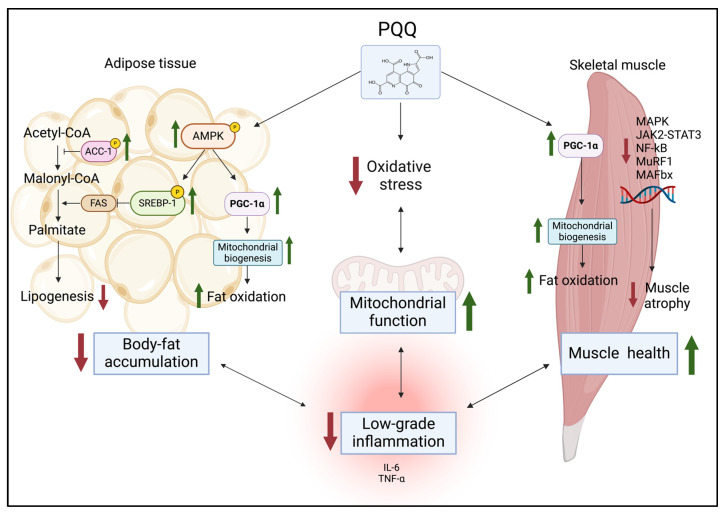
Representation of the main pathways involved in the multiple effects of PQQ on adipose metabolism, muscle health, and mitochondrial function. Up green arrows indicate an increase while down red arrows a decrease. Truncated black arrows suggest inhibitory activity. ACC, acetyl-CoA carboxylase; AMPK, AMP-activated protein kinase; FAS, fatty acid synthase; JAK, Janus kinase; MAFbx, muscle atrophy F-box; MAPK, mitogen-activated protein kinase; MuRF-1, muscle-specific RING finger-1; NF-κB, nuclear factor kappa-light-chain-enhancer of activated B cells; PGC-1α, proliferator-activated receptor gamma coactivator 1-alpha; SREBP, sterol regulatory element-binding protein 1. Created with BioRender.com, accessed on 31 July 2024.

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
