# Peer review of "Metabolic and Biochemical Effects of Pyrroloquinoline Quinone (PQQ) on Inflammation and Mitochondrial Dysfunction: Potential Health Benefits in Obesity and Future Perspectives"

_antioxidants, 2024, doi:10.3390/antiox13091027_

Round 1
Reviewer 1 Report
Major concern:
1. There have been many study reports on the nutritional and health-care functions of PQQ, which mainly be due to the antioxidant and anti-inflammatory effects of PQQ. However, whether PQQ can directly enter adipose tissue or muscle tissue and affect the metabolism in the tissues is unknown, so the title of this study is inappropriate and should be changed.
2. In addition to collecting and reviewing existing reports, a good literature review should also propose directions and entry points for future research, rather than just emphasizing that human trials are insufficient.
3. There are many typing errors in this study.
4. PQQ exists in many foods but in very trace amounts. The authors should explain the appropriate amount of food from the perspective of food and nutrition.
There are many typing erroes. English check of the manuscript should be performed by native English-speaking professional before re-submit the manuscript.
Author Response
We would like to thank the editor for the support and the reviewers for their insightful comments and suggestions. We have carefully considered the constructive recommendations made by the reviewers and modified the manuscript accordingly. We have incorporated almost all the suggestions made by reviewers highlighted in yellow in the text; we also provided a thorough, one-by-one answer to each comment. We believe our revised manuscript addresses the reviewer concerns that helped to significantly improve the quality of our paper. We look forward to receiving your feedback.
Reviewer 1
General comment: “The title should be changed”
Response: Title changed
General comment: There are only two figures in this study. It should be added more.
Response: We uploaded a new figure (Figure 1) illustrating the main pathways regulated by PQQ.
General comment: The conclusion can't support the conclusion.
Response: The updated version includes more details at the bottom of each paragraph, suggesting potential areas for future investigation, with particular emphasis on the gaps in the current literature. In addition, conclusions briefly summarize the main literature gaps and also include tips to design new high-quality trials.
Comment 1: There have been many study reports on the nutritional and health-care functions of PQQ, which mainly be due to the antioxidant and anti-inflammatory effects of PQQ. However, whether PQQ can directly enter adipose tissue or muscle tissue and affect the metabolism in the tissues is unknown, so the title of this study is inappropriate and should be changed.
Response 1: Thanks for noticing that. The title has been changed. Furthermore, as you pointed out, it remains unclear whether PQQ can directly enter adipose or muscle tissue and influence the associated metabolism. In light of that, we have also deeply improved the manuscript and focused on these crucial points.
See lines 190-193 and 375-378.
Comment 2: In addition to collecting and reviewing existing reports, a good literature review should also propose directions and entry points for future research, rather than just emphasizing that human trials are insufficient.
Response 2: We appreciate your suggestion to include directions for future research in our review. The updated version includes more details at the bottom of each paragraph, suggesting potential areas for future investigation, with particular emphasis on the gaps in the current literature.
See line 190-194, 244-249, 312-314 and 375-378.
In addition, conclusions briefly summarize the main literature gaps and include tips to design new high-quality trials.
See lines 400-410.
Comment 3: There are many typing errors in this study.
Response 3: Thanks for noticing that. We have thoroughly revised the manuscript and identified errors to ensure clarity and precision.
Comment 4: PQQ exists in many foods but in very trace amounts. The authors should explain the appropriate amount of food from the perspective of food and nutrition.
Response 4: Thanks for your intriguing observation. In line with that, we have added a detailed explanation about the amounts of PQQ from a nutrition perspective.
See lines 71-74 and 80-84.

Reviewer 2 Report
The manuscript has an interesting idea, addressing the metabolic and biochemical implications of PQQ and its potential health benefits in obesity.
One of the striking characteristics of this manuscript is the imbalance between the body of the manuscript (6 ½ pages, including 2 figures) and the bibliography section (51/2 pages). 4 of 84 citations are for 2024 published articles, 5 from 2023, 4 from 2022).
The structure and antioxidant effect of PQQ should be illustrated by a corresponding image< the same observation applies to all the cellular/biochemical pathways regulated by PQQ (page 2 lines 89-99)
Authors state “For example, in obese mice, PQQ supplemen-122 tation decreased total body fat and visceral fat compared to the control groups”- the level of exposure should be indicated (dose/duration of treatment). Also, for every example, the exact tpe of study model should be described – animal type/cell type/ treatment, etc. This applies to all animal/human studies presented in the manuscript, especially since in some parts of the manuscript authors already included the doses used in preclinical experiments.
Authors state “However, a decrease in mean LDL-cholesterol lev-154 els (from 136.1 to 127.0 mg/dL) was observed in the PQQ group, indicating a potential 155 effect of PQQ on hypercholesterolemia and hypertension [58].”- was the change statistically significant? The effects on blood pressure should be backed by data., In the current form, the hypercholesterolemia and the effect on hypertension are not justified.
“Moreover, it has 174 been observed that adding PQQ (at doses of 10-30ug for 24-48 hours) in liver cells of defi-175 cient-PQQ mice…”- the model should be properly described. How is PQQ added to liver cells of deficient mice?
“In a model of TNF- α-induced 179 mitochondrial damage, PQQ supplementation (at a dose of 10 μmol/L) reduced the mito-180 chondrial damage” – what is the exact experimental model for this effect?
“In septic rats, PQQ supplementation alleviated acute liver injury and apoptosis of 233 liver cells [19]”- dose??? Doses should also be indicated at lines 239-241 as well as 243-246, 249-250.
The language should be revised for small errors – one example bellow
“ These factors promote energy 44 dysregulation and interfere with skeletal muscle mass protein metabolism ???by enhancing…”
Author Response
Response to Reviewer 2 Comments
We would like to thank the editor for the support and the reviewers for their insightful comments and suggestions. We have carefully considered the constructive recommendations made by the reviewers and modified the manuscript accordingly. We have incorporated almost all the suggestions made by reviewers highlighted in yellow in the text; we also provided a thorough, one-by-one answer to each comment. We believe our revised manuscript addresses the reviewer concerns that helped to significantly improve the quality of our paper. We look forward to receiving your feedback.
General Comments:
The manuscript has an interesting idea, addressing the metabolic and biochemical implications of PQQ and its potential health benefits in obesity.
One of the striking characteristics of this manuscript is the imbalance between the body of the manuscript (6 ½ pages, including 2 figures) and the bibliography section (51/2 pages). 4 of 84 citations are for 2024 published articles, 5 from 2023, 4 from 2022).
Response:
Thanks for your positive feedback regarding the imbalance between the length of the manuscript body and the bibliography section. The revised version now incorporates additional details, increased text, and a new figure. In addition, although new references have been added, other unnecessary ones have been removed.
Detailed Comments:
Comment 1: The structure and antioxidant effect of PQQ should be illustrated by a corresponding image< the same observation applies to all the cellular/biochemical pathways regulated by PQQ (page 2, lines 89-99).
Response 1: Thanks for the comment. We uploaded a new figure (Figure 1) illustrating the main pathways regulated by PQQ.
Comment 2: Authors state “For example, in obese mice, PQQ supplementation decreased total body fat and visceral fat compared to the control groups”- the level of exposure should be indicated (dose/duration of treatment). Also, for every example, the exact type of study model should be described – animal type/cell type/ treatment, etc. This applies to all animal/human studies presented in the manuscript, especially since in some parts of the manuscript authors already included the doses used in preclinical experiments.
Response 2: Thanks for your accurate feedback. We agree that providing these details is crucial for a better interpretation of the findings. In line with that, we have revised the entire manuscript to include the specific doses, durations of treatment, and types of study models cited. All changes are highlighted in yellow.
Comment 3: Authors state “However, a decrease in mean LDL-cholesterol levels (from 136.1 to 127.0 mg/dL) was observed in the PQQ group, indicating a potential effect of PQQ on hypercholesterolemia and hypertension [58].”- was the change statistically significant? The effects on blood pressure should be backed by data., In the current form, the hypercholesterolemia and the effect on hypertension are not justified.
Response 3: Thanks for your insightful comment. We checked the cited study and clarified whether the observed decrease in LDL-cholesterol levels was statistically significant. Furthermore, we have deleted the effect of PQQ on blood pressure in the absence of significant changes.
See lines 182-187.
Comment 4: “Moreover, it has been observed that adding PQQ (at doses of 10-30mg for 24-48 hours) in liver cells of deficient-PQQ mice…”- the model should be properly described. How is PQQ added to liver cells of deficient mice?
Response 4: Thanks for the observation. We concur that a clear outline of the methodology would enhance understanding. In response to your comment, we have revised the manuscript to provide a more detailed description of the model. See lines 208- 213
Comment 5: “In a model of TNF- α-induced mitochondrial damage, PQQ supplementation (at a dose of 10 μmol/L) reduced the mitochondrial damage” – what is the exact experimental model for this effect?
Response 5: Thanks for the comment. The updated version includes a more detailed description of the experimental model. See lines 215-219.
Comment 6: “In septic rats, PQQ supplementation alleviated acute liver injury and apoptosis of liver cells [19]”- dose??? Doses should also be indicated at lines 239-241 as well as 243-246, 249-250.
Response 6: Thanks for your precious feedback. We have now included the specific doses of PQQ supplementation in the relevant sections. See lines 278-282, 290, 299,303,304, 306, 307.
Comment 7: The language should be revised for small errors – one example bellow. “ These factors promote energy dysregulation and interfere with skeletal muscle mass protein metabolism ???by enhancing…”
Response 7: Thanks for noting that. The updated version has been deeply improved and further revised in order to enhance the quality of the manuscript.

Round 2
Reviewer 1 Report
The manuscript is now suitable for publication.
The manuscript is now suitable for publication.
Reviewer 2 Report
Authors responded to previous questions and this contributed to an improvement of the manuscript that is, in the current form, acceptable for publication
Authors added the figure that is relevant to the manuscript and also modified the body of the manuscript according to suggestions, thus correcting the imbalance with the bibliography list